# Unraveling the Signaling Dynamics of Small Extracellular Vesicles in Cardiac Diseases

**DOI:** 10.3390/cells13030265

**Published:** 2024-01-31

**Authors:** Sheila Caño-Carrillo, Juan Manuel Castillo-Casas, Diego Franco, Estefanía Lozano-Velasco

**Affiliations:** Cardiovascular Development Group, Department of Experimental Biology, University of Jaén, 23071 Jaén, Spain; scano@ujaen.es (S.C.-C.); jmcasas@ujaen.es (J.M.C.-C.); dfranco@ujaen.es (D.F.)

**Keywords:** intercellular communication, extracellular vesicles, cardiovascular diseases, biomarkers, prognosis, diagnosis

## Abstract

Effective intercellular communication is essential for cellular and tissue balance maintenance and response to challenges. Cellular communication methods involve direct cell contact or the release of biological molecules to cover short and long distances. However, a recent discovery in this communication network is the involvement of extracellular vesicles that host biological contents such as proteins, nucleic acids, and lipids, influencing neighboring cells. These extracellular vesicles are found in body fluids; thus, they are considered as potential disease biomarkers. Cardiovascular diseases are significant contributors to global morbidity and mortality, encompassing conditions such as ischemic heart disease, cardiomyopathies, electrical heart diseases, and heart failure. Recent studies reveal the release of extracellular vesicles by cardiovascular cells, influencing normal cardiac function and structure. However, under pathological conditions, extracellular vesicles composition changes, contributing to the development of cardiovascular diseases. Investigating the loading of molecular cargo in these extracellular vesicles is essential for understanding their role in disease development. This review consolidates the latest insights into the role of extracellular vesicles in diagnosis and prognosis of cardiovascular diseases, exploring the potential applications of extracellular vesicles in personalized therapies, shedding light on the evolving landscape of cardiovascular medicine.

## 1. Introduction

Intercellular communication is an essential activity for the maintenance of cellular and tissue homeostasis as well as responding to pathological processes. For this purpose, cells have different mechanisms depending on the distance of cell-to-cell communication requirements. Short cellular cross talk is mediated by direct cell-to-cell contact or secretion of soluble factors, whereas long range communication is driven through ligand–receptor interactions such as cytokine and/or hormone release [1,2]. However, in recent decades a novel mechanism of communication has emerged, where extracellular vesicles (EVs) are implicated [2]. EVs are constitutively produced by most of the cell types and consist in a lipidic bilayer membrane that encloses biological contents derived from the original cell, i.e., proteins, nucleic acids and lipids that can alter the biology of the distant cell [3,4]. The interest in EV biology has grown exponentially and many different types of EVs have been classified depending on their size, biogenesis and their biophysical properties [5,6] (Figure 1). There are two primary types of EVs, exosomes and ectosomes, whose classification is determined by their formation process. The genesis of these vesicles is contingent upon the assembly of local microdomains within endocytic membranes for exosomes and the plasma membrane for ectosomes, the latter subclassified into microvesicles and apoptotic bodies [7,8].

Exosomes are small EVs ranging from 30 to 100 nm. They are formed by inward budding of the early endosome membrane which subsequently matures into multivesicular bodies (MVBs) [9,10,11,12]. MVBs are involved in protein sorting, recycling, storage, transport and exosome release into the extracellular space [11,13,14,15,16]. Exosomes were originally considered as a cellular mechanism to eliminate cell debris; however, since few years ago, they are considered as principal mediators of cell-to-cell communication, cell and tissue maintenance and disease progression [17]. Moreover, exosomes can act as antigen-presenting vesicles participating in the stimulation of immune response [18,19]. Microvesicles (MVs) are EVs with a range from 200 nm to 1 µm which are directly shed as an outward budding of the plasma membrane [10,11,12,13,20,21]. Similarly to exosomes, MVs are involved in cell-to-cell communication between healthy and diseased cells [22,23]. Finally, apoptotic bodies are released into the extracellular space as blebs of cells undergoing apoptosis, and they have been reported as the bigger EVs, with a size range from 1 to 4 µm [13,24]. These apoptotic bodies are formed when the cytoskeleton is separated from the cellular plasma membrane due to increased hydrostatic pressure after the cell contracts [25]. The composition of the apoptotic bodies are organelles, chromatin and glycosylated proteins, in contrast to exosomes and MVs [13,26,27] (Figure 1).

As mentioned previously, most cell types release EVs of different sizes, composition and subcellular origin, and moreover, these EVs can be found in different body fluids such as plasma, saliva and urine [28,29,30]. Therefore, due to their presence in liquid biopsies, EVs are considered as potential disease biomarkers. Understanding the mechanisms by which molecular cargos are loaded into the EVs will provide the key to completely understanding the role of EVs in cellular communication during disease development and progression [31,32,33,34] (Figure 1).

Cardiovascular disease (CVD) is one of the leading causes of morbidity and mortality around the world [35]. CVDs cover a wide range of pathological conditions where the most common types are ischemic heart disease, structural cardiomyopathies, electrical heart diseases and finally heart failure (HF) [36]. There have been major steps forward in diagnosis and prognosis of CVDs that have improved patients’ survival and their quality of life, but there is still much work to do for CVD prevention and palliation, given the continued high mortality rate [37]. Several studies have recently indicated that small EVs can be released by cardiovascular cells, such as cardiomyocytes (CM), endothelial cells (EC), cardiac fibroblasts (CF), platelets, smooth muscle cells (SMCs), leucocytes, monocytes and macrophages [38]. These small EVs play important biological roles in the maintenance of normal cardiac structure and function, whereas small EVs are able to change their composition under pathological conditions, thus contributing to the development of CVDs [39,40,41].

In this review, we summarize the current state-of-the-art small EVs in the prognosis and diagnosis of CVDs, as well as the possible applications of small EVs in personalized therapies.

## 2. Small EVs in the Pathological Process of the Myocardial Infarction

Myocardial infarction (MI) is one of the most common CVDs, representing the main leading cause of death in the world [42]. Pathologically, MI is defined as the death of CMs due to a prolonged lack of oxygen in a specific area of the myocardium, initiating an apoptotic process leading to necrosis of the cardiac muscle and subsequently to other cardiac diseases such as arrhythmias or HF [43,44,45,46] (Figure 2A). The main etiology of MI is associated with the rupture of the atherosclerotic plate [47], although other causes, such as coronary artery embolism or coronary vasospasm have been also described [48]. After MI and the consequent death of CMs, autophagy and inflammatory processes begin as a strategy to remove the damaged tissue, allowing the replacement of the necrotic area with fibrotic tissue [46,49,50,51,52,53,54]. Furthermore, mechanisms of revascularization are also activated despite the low regenerative capacity of the heart after cardiac injury [55,56]. Currently, several studies show that exosomes play an important role in post-MI processes and participate in cellular communication regulating cardiac remodeling after MI [57,58,59]. For this reason, many authors focus their attention on the main mechanisms that regulate the production and content of these exosomes using transcriptomic and proteomic approaches, highlighting their detection as biomarkers after MI and thus their plausible therapeutic use [60,61,62]. The following paragraphs summarize the most important features of these exosomes in the MI context.

### 2.1. Small Extracellular Vesicle Transcriptomic Analyses in Myocardial Infarction

The discovery of new functions of exosomes has allowed the use of these small EVs as a potential tool to identify molecules that can be used as biomarkers after MI [63]. Currently, the main diagnostic biomarker for MI is cardiac troponin (T/I) [64], even though other more sensitive biomarkers have emerged, for example, non-coding RNAs, which allow for an earlier diagnosis [65,66]. Within non-coding RNAs, an increasing number of transcriptomic studies identified miRNAs in exosomes after MI [67,68,69]. Some analyses of circulating exosomes in the serum of MI patients have revealed a large number of miRNAs that are deregulated. For example, miR-203, miR-4516, or miR-183, which regulate the activity of several protein kinases in CMs, are upregulated, confirming their identification as MI biomarkers [67,70]. Another transcriptomic study identifies the deregulation of miRNA levels in exosomes on a large scale, identifying around 500 upregulated and downregulated miRNAs in MI exosomes, highlighting miR-6718 and miR-4329 [68]. Guo et al. (2021) reveal that, in addition to differential expression in exosomal miRNAs identifying up to 18 miRNAs as biomarkers, exosomes differ in size, being smaller in MI patients [69] (Table 1).

Some miRNAs have been widely studied due to their essential role in post-MI inflammatory, fibrotic and angiogenic processes, i.e., miR-126 or miR-155, which are upregulated in exosomes of MI patients, and miR-21 or miR-146a-5p, which are downregulated [71,72,73]. Specifically, miR-146a-5p is associated with inflammation after MI by regulating M1 macrophage polarization through the regulation of TNF Receptor-Associated Factor 6 (TRAF6) [74]. Regarding angiogenesis, another transcriptomic analysis of exosomes derived from MI patients revealed up to 40 differentially expressed miRNAs, highlighting the downregulation of miR-143. In vitro assays demonstrated that this downregulation promotes angiogenesis through the insulin-like growth factor 1 receptor and nitric oxide (IGF-IR/NO) signaling pathway [75]. In contrast, in vitro assays in CMs after ischemia/reperfusion (I/R) describe exosomes with high levels of miR-143 and miR-222, both enhancing angiogenesis and cardiac remodeling [76]. In addition, in vitro and in vivo assays in necrotic dendritic cells revealed the presence of eight upregulated miRNAs related to the regulation of angiogenesis, in which miR-494 is the most significant due its role in promoting revascularization after injury [77]. A similar result was demonstrated by Duan et al. (2022), where the analysis of exosomes extracted from the peripheral serum of patients with MI identified high levels of miR-126-3p, which promotes angiogenesis through regulation of the miR-126-3p/TSC1/mTORC1/HIF-1α signaling pathway [78] (Table 1).

The protective role of some exosomal miRNAs was also confirmed in ferroptosis after MI. Low levels of miR-26b-5p were described in exosomes derived from MI patients, which can reduce ferroptosis by positively regulating Solute Carrier Family 7 Member 11 (SLC7A11) [79]. Finally, other studies also associate miRNA transport in exosomes derived from MI patients with subsequent apoptosis [80]. Sun et al. (2022) revealed 52 differentially expressed miRNAs by transcriptomic analysis, highlighting the upregulation of miR-133a-3p, miR-151a-5p, miR-199b-5p, miR-374b-5p, miR-503-5p, and miR-708-5p. Concretely, they focused on miR-503-5p because of its ability to promote apoptosis through the regulation of Peroxisome Proliferator-activated Receptor Gamma Coactivator-1β (Ppargc-1β) and Sirtuin 3 (SIRT3) in CMs [80] (Table 1).

Although miRNAs are the most frequent non-coding RNAs studied in exosomes as MI biomarkers, long non-coding RNAs (lncRNAs) and circular RNAs (circRNAs) are also analyzed. For lncRNAs, sequencing profiles allowed us to identify 518 differentially expressed in post-MI exosomes, highlighting ENST00000556899.1 and ENST00000575985.1, which are upregulated, thus supporting their involvement in the regulation of post-MI processes [81]. More specifically, other lncRNAs like TUG1 or HCG15 have also been detected as being upregulated in MI exosomes, and in vitro and in vivo assays show that these lncRNAs are implicated in the inhibition of angiogenesis and cell viability through the regulation of HIF-1α/VEGF-α and NF-κβ/p65, respectively [82,83]. Despite being smaller in number, circular RNAs have been reported to have a significant advantage over linear RNAs because they exhibit better stability and can therefore be identified as biomarkers of MI for a longer period of time [84]. For this reason, some transcriptomic studies focused on the analysis of circRNAs in MI exosomes, highlighting, for example, the upregulation of circ_0020887 and circ_0009590 in the exosomes patients with ST segment elevation on electrocardiogram [85]. Another circRNA that also increases its expression in exosomes after MI is circITGB1. Within in vivo assays using a mouse model, these exosomes activate dendritic cells and exacerbate cardiac damage and inflammation through the miR-342-3p/NFAM1 pathway [86]. In vitro assays have also been reported in CMs subjected to hypoxia where high levels of circ_HIPK3 and circ_SLC8A1 are detected [87,88]. These circ_HIPK3-loaded exosomes target cardiac microvascular endothelial cells (CMVECs), protecting them from oxidative stress through the regulation of the miR-29a/VEGFA and miR-33a-5p/IRS1 pathways [89,90]. On the other hand, high levels of exosomal circSLC8A1 promote the inflammatory process and oxidative stress in other CMs, leading to their apoptosis through the regulation of the miR-214-5p/TEAD1 axis [88] (Table 1).

**Table 1 cells-13-00265-t001:** Summary of ncRNAs associated with myocardial infarction (↑ upregulated and ↓ downregulated).

Assay	DE ncRNAs	Main ncRNAs	Sample	Ref.
**miRNAs**				
Liu et al. (2023)	↑ miR-4516 ↑ miR-203	miR-4516, miR-203	Plasma (MI patient)	[67]
Zhao et al. (2019)	85 miRNAs	miR-183, miR-92, miR-4709, miR-550, miR-223	Plasma (MI patient)	[70]
Chen et al. (2021)	↑ 544 miRNAs ↓ 518 miRNAs	miR-6718, miR-4329	Serum (MI patient)	[68]
Guo et al. (2021)	↑ 138 miRNAs ↓ 208 miRNAs	miR-143-3p, miR-23b-5p, miR-106b-5p, miR-33a-5p	Plasma (MI patient)	[69]
Ling et al. (2020)	↑ miR-126 ↓ miR-21	miR-126 miR-21	Serum (MI patient)	[71]
Geng et al. (2020)	↑ 20 miRNAs ↓ 20 miRNAs	miR-143	Serum (MI patient)	[75]
Ribeiro-Rodrigues et al. (2017)	↑ 3 miRNAs ↓ 4 miRNAs	miR-143, miR-222	H9c2 primary CMs	[76]
Liu et al. (2020)	↑ 8 miRNAs	miR-494-3p	MI mouse model	[77]
Sun et al. (2022)	↑ 12 miRNAs ↓ 11 miRNAs	miR-503	Peripheral blood MI mouse model	[80]
**LncRNAs**				
Zheng et al. (2020)	↑ 245 lncRNAs ↓ 273 lncRNAs	ENST00000556899.1 ENST00000575985.1	Plasma (MI patient)	[81]
Dang et al. (2023)	↑ TUG1	TUG1	Plasma (MI patient) MI mouse model	[82]
Lin et al. (2021)	↑ 29 lncRNAs ↓ 36 lncRNAs	HCG15	Serum (MI patient)	[83]
**circRNAs**				
Wang et al. (2023)	428 circRNAs	circ_0020887, circ_0009590	Plasma (MI patient)	[85]
Zhu et al. (2022)	↑ 10 circRNAs ↓ 10 circRNAs	circ_ITGB1, circ_SLC7A1, circ_ATG5, circ_POLR1A	Plasma (MI patient)	[86]

### 2.2. Small Extracellular Vesicle Proteomic Analyses in Myocardial Infarction

Just as many transcriptomic studies have identified essential biomarkers for MI; there are also proteomic analyses that reveal the presence of distinct proteins in post-MI exosomes that can be used as biomarkers for MI diagnosis [91]. Xie et al. (2022) analyzed the protein profile within the exosomes of post-MI patients, identifying 72 differentially expressed proteins. Notably, three proteins exhibited elevated levels: Plasminogen (PLG), Complement Component C8 Beta (C8B) and Thrombin (F2) [91]. Furthermore, additional research emphasizes the presence of proteins within these exosomes that are involved in the inflammatory process and cardiac remodeling, such as Phosphatase and Tensin Homolog (PTEN) or Matrix Metalloproteinase-9 (MMP-9) [71,92]. In the post-fibrotic process after MI, some proteins present in exosomes have also been described, such as the transcriptional cofactor Limb Bud And Heart Development Protein Homolog (LBH). These exosomes produced by damaged CMs are taken up by CFs and activate Crystallin Alpha B (CRYAB), promoting further proliferation of CFs and their differentiation into myofibroblasts [59]. Post-MI exosomes have also been reported to carry proteins that promote cardiac repair and remodeling after damage, for example, Clusterin or Profilin 2 (PFN2) [93,94]. The levels of PFN2 are elevated in exosomes produced by ECs following MI. In both in vitro and in vivo assays, these exosomes demonstrate an increase in angiogenesis and cardiac improvement after damage through regulation of the PI3K/PNF2/ERK axis [94].

Finally, other studies highlight the importance of exosomes as biomarkers for MI diagnosis. This research conducted a proteomic study by comparing plasma from control and MI patients and analyzing the protein profile in post-MI exosomes. These results revealed 11 proteins that were deregulated in the exosomes compared to control plasma. However, three of these proteins, Chymotrypsin C (CTRC), Proto-oncogene Tyrosine-protein Kinase SRC (SRC), and C-C Motif Chemokine Ligand 17 (CCL17), did not exhibit downregulation when comparing their levels in serum between MI and control patients. Therefore, this analysis justified the need to also analyze post-MI exosomes to obtain additional diagnostic information that are not available only from plasma samples [95].

### 2.3. Mechanistic Insights into Small Extracellular Vesicle Related with Myocardial Infarction

As discussed in previous paragraphs, exosomes are involved in the regulation of the activation/inhibition of the main processes that occur after MI, such as inflammation, fibrosis, angiogenesis or apoptosis, as well as allowing communication between cells to coordinate these processes [57,58,59]. After MI, M1 macrophages are activated and increase the production of exosomes (M1-exos) with miR-155, which deliver this miRNA to CFs, thereby enhancing the inflammatory and fibrotic process [96]. M1-exos with high levels of miR-155 are also implicated in the inhibition of angiogenesis since these M1-exos can also be transferred to ECs and reduce angiogenesis via Sirt1/AMPKα2 and RAC1/PAK2 signaling after MI [97]. Macrophages are not the only cells that can release exosomes carrying miRNAs involved in post-MI process regulation. In vivo assays in a mouse MI model show that after MI, T CD4^+^ cells are activated and promote the synthesis of exosomes in which miR-142-3p is upregulated and promotes myofibroblast differentiation in ECs via miR-142-3p/APC/Wnt signaling [98]. Exosomes are involved in paracrine regulatory processes, as damaged CMs release exosomes after MI that will act on neighboring cells such as other CMs or ECs. This exosomal function is confirmed by the study of Gou et al. (2016), which showed that infarcted CMs generate exosomes that exhibit highly expressed miR-19a-3p, which are delivered to ECs and inhibit angiogenesis regulating Hypoxia-inducible factor 1-alpha (HIF-1α) [99]. Furthermore, post-MI CMs can generate exosomes after cardiac damage carrying miR-328-3p, which increase the apoptotic process through upregulation of Caspase 3 (Casp3) [100]. CFs also receive these exosomes produced by the CMs after MI, increasing their proliferation and the myofibroblast differentiation due to the high levels of miRNAs such us miR-208 or miR-92a [101,102,103]. Cell-to-cell communication via exosomes also takes place in the opposite direction, i.e., ECs can generate exosomes carrying miR-503 and promote CM apoptosis after MI [80]. Another example of miRNA that can be transferred between CMs after MI is miR-30a, whose levels are upregulated in exosomes extracted from the serum of MI patients. In vitro assays show that after hypoxia, these exosomes regulate the autophagy process between hypoxic CMs by upregulating genes such as Beclin-1 (BECN1), Autophagy Related 12 (Atg12), or Microtubule Associated Protein 1 Light Chain 3 Alpha (LC3I/II) [104]. Exosomes, carrying low levels of miR-342-3p as a consequence of MI, contribute to the regulation of both autophagy and apoptosis. Both in vitro and in vivo assays reveal that miR-342-3p regulates apoptosis and autophagy through the SRY-Box Transcription Factor 6 (SOX6) and Transcription Factor EB (TFEB), respectively [105]. A similar process, ferroptosis, also plays a relevant role in exosome regulation in post-MI processes. After MI, ferroptotic CMs release exosomes with low levels of miR-106b-3p, which promotes the activation of the Wnt pathway and increases the polarization of M1 macrophages, enhancing the inflammatory process after damage [106].

Other authors focused on the differential expression of proteins involved in the biogenesis, uptake or polarization of exosomes generated after MI. One example is the high level of CD44 in MI, which is involved in the synthesis of exosomes after MI through the positive regulation of Fibroblast Growth Factor Receptor 2 (FGFR2), as well as the subsequent uptake of these exosomes by ECs [107]. Regarding the ability of exosomes to migrate and the factors that trigger their polarization to the region of interest, it has been reported that dendritic cells increase the expression of C-C Motif Chemokine Receptor 7 (CCR7) after damage, and they generate exosomes containing both CCR7 and its ligands. These exosomes can target the spleen to activate CD4+ T cells, which produce anti-inflammatory cytokines that promote cardiac remodeling [108,109]. In addition, an external inflammatory stimulus, such as a decrease in the anti-inflammatory cytokine IL-10, can modify the protein content of exosomes. One protein that modifies its levels in exosomes is Integrin Linked Kinase (ILK), whose high levels promote the activation of NF-κβ, enhancing the inflammatory response and decreasing angiogenesis in the cells that receive these exosomes [110,111].

### 2.4. Therapeutic Approaches

Although many studies describe the potential use of exosomes as biomarkers for the diagnosis of MI, several studies additionally analyzed the use of exosomes as tools to transport specific molecules that promote cardiac repair after MI, thus becoming a widely used strategy in recent years [112,113,114,115]. The most frequent strategy is based on the use of exosomes derived from different mesenchymal stem cells (MSCs) that carry a specific molecule (RNA or protein) involved in the processes of inflammation, fibrosis or angiogenesis, among others [116,117,118].

Currently, there are numerous studies that use exosomes derived from different cell types to transport a specific non-coding RNA to the damaged area after MI (Table 2). One of the most widely used miRNAs as a therapeutic tool due to its role in inflammation and fibrosis is miR-21. Several studies extract exosomes derived from different cell types such as MSCs, cardiac telocytes (CTs), or even serum from control individuals loaded with miR-21, using these exosomes as a treatment for MI in both in vitro and in vivo assays [119,120,121]. Administration of these miR-21-loaded exosomes shows an improvement in cardiac function after MI due to the positive regulation of the angiogenic process and the inhibition of CM apoptosis and fibrosis through the regulation of PTEN and p53/Cdip1/Casp3 signaling pathways, among others [119,121,122,123] (Figure 2A). MSC-derived exosomes, both under control and hypoxic conditions and carrying high levels of miR-125b, have also been used as a therapy against CM apoptosis after MI [124,125]. One of the most commonly used types of MSCs to obtain exosomes, administered as a treatment for MI, are those derived from human umbilical cord mesenchymal stem cells (HUCMSCs) [126,127]. These cells have been used to obtain exosomes that act as a vehicle to transport miRNAs such as miR-23, miR-133 or miR-223 to the infarcted area, which promote cardiac repair by activating angiogenesis and reducing inflammation, ferroptosis or fibrosis [128,129,130] (Figure 2). Other studies use this strategy to produce exosomes derived from cardiosphere-derived cells (CDCs) or MSCs which carry miR-181, regulating macrophage polarization and reducing inflammation when they are administered after MI [131,132]. There is also evidence that the therapeutic use of MSC-derived exosomes carrying specific proteins such as Itchy E3 Ubiquitin Protein Ligase (ITCH), Fibronectin Type III Domain Containing 5 (FNDC5) or Stromal Cell-Derived Factor 1 (SDF1) promote cardiac repair after their administration [133,134,135] (Figure 2A) (Table 2).

Another treatment strategy post MI involves the modification of gene expression in the cells from which exosomes are subsequently extracted. This modification leads to changes in the content of these exosomes, offering a targeted approach for therapeutic interventions [136,137,138]. Several studies report that the overexpression of genes such as Hypoxia Inducible Factor 1 Subunit Alpha (HIF1-α) in MSCs results in a modification of the content of these exosomes, making them able to ameliorate cardiac damage after MI by promoting angiogenesis and reducing fibrosis, apoptosis or inflammation [139,140]. Overexpression of GATA-Binding Protein 4 (GATA4) in the cells from which exosomes are extracted has also been widely used to modify the exosome content for use as a treatment after MI [141,142,143]. He et al. (2018) revealed that overexpression of GATA4 in bone marrow mesenchymal stem cells (BMSCs) results in the production of exosomes that enhance myocyte precursor differentiation and reduce apoptosis after MI. This effect is due to a shift in the protein pattern carried by these exosomes, in which eight proteins associated with differentiation and six related to apoptosis have been identified with differential expression [141]. Finally, the reparative capacity of neonatal serum-derived exosomes has also been reported, identifying up to 28 ligands that can promote angiogenesis when these exosomes are administered after MI [144] (Table 2). Therefore, the understanding of molecular and structural aspects of exosomes has paved the development of a promising tool for MI treatment.

**Table 2 cells-13-00265-t002:** Summary of ncRNAs for therapeutically approaches in myocardial infarction (↑ upregulated and ↓ downregulated).

**miRNAs**						
**Source of Exosomes**	**In Vitro** **Essay**	**In Vivo Model**	**miRNA**	**Signaling**	**Biological Effects**	**Refs.**
MSCs BMSCs	H9c2 HUVECs	MI (LAD)	miR-210	AIFM3/PI3K/AKT AIFM3/p53 Ephrin, Casp8	↑ CMs viability ↓ Fibrosis, apoptosis	[145,146,147,148]
ADSCs	-	MI (LAD)	miR-205	-	↑ Angiogenesis ↓ CM apoptosis, cardiac fibrosis	[149]
MSCs BMSCs	NMCMs	MI (LAD)	miR-125-5p miR-125b-5p	p53/BAK1 p53/Bnip3	↑ Cardiac function ↓ CM apoptosis, autophagy	[124,125]
CTs MSCs plasma BMSCs	CMECs H9c2 HUVECs MRCMs	MI (LAD)	miR-21-5p miR-21 miR-21a-5p	p53/Cdip1/Casp3 BTG2 PDCD4 PTEN	↑ Angiogenesis, CM proliferation ↓ Apoptosis, fibrosis	[119,120,122]
MSCs	CMs	MI (LAD)	miR-25-3p	FASL/PTEN EZH2/H3K27me3	↓ Apoptosis	[150]
EPDCs	H9c2 CMs	MI (LAD)	miR-27a miR-100 miR-30a miR-30c	-	↑ Proliferation	[151]
MSCs	ECs H9c2	-	miR-153-3p	Angpt1/Vegf/ Vegfr2/PI3k/Akt/ eNOs	↑ Angiogenesis ↓ Apoptosis	[152]
ADSCs	HMVEC	MI (LAD)	miR-31	FIH1/HIF-1α	↑ Angiogenesis	[153]
CDCs HUCMSCs	Macrophages PBMCs	MI (LAD)	miR-181b miR-181a	PKCδ c-Fos	↑ Macrophague polarization, cardiac protection ↓ Inflammation	[131,132]
BMSCs	H9c2	MI (LAD)	miR-143-3p	CHK2/Beclin2	↓ Apoptosis, autophagy	[154]
ADSCs	H9c2	MI (LAD)	miR-93-5p	Atg7/TLR4	↑ Cardiac protection ↓ Inflammation	[155]
Serum ADSCs	HUVECs H9c2	MI (LAD)	miR-126-3p miR-126	TSC1/mTORC1/ HIF-1α	↑ Angiogenesis ↓ Inflammation, fibrosis	[78,156]
BMSCs	H9c2	MI (LAD)	miR-338	MAP3K2/JNK	↑ Cardiac function ↓ Apoptosis	[157]
MSCs	HUVECs	MI (LAD)	miR-132	RASA1	↑ Angiogenesis	[126]
BMSCs	-	MI	miR-29b-3p	ADAMTS16	↑ Angiogenesis ↓ Fibrosis	[158]
ADSCs	CMs	MI (LAD)	miR-671	TGFBR2/Smad2	↓ Fibrosis, inflammation, apoptosis	[159]
BMSCs HUCMSCs HUVECs	H9c2	MI (LAD)	miR-24 miR-24-3p	Plcb3/NF-κβ CCR2	↑ M2 macrophague polarization ↓ CM apoptosis	[160,161]
M2 macrophage	-	MI (LAD)	miR-1271-5p	SOX6	↓ Apoptosis	[162]
BMSCs	H9c2	MI (LAD)	miR-30e	LOX1/NF-κβ-p65/Casp9	↓ Apoptosis, fibrosis	[163]
EPCs	CFs	MI (LCA)	miR-1246 miR-1290	EFL5/CD31/ VEGFR2/α-SMA SP1/CD31/VEGFR2/ α-SMA	↑ Angiogenesis ↓ Fibrosis	[164]
HUCMSCs	CMs HUVECs	MI (LAD)	miR-214-3p	PTEN/AKT	↑ Angiogenesis ↓ Apoptosis	[165]
ADSCs CMs	H9c2 M1 macrophages	MI (LAD)	miR-146-a	EGR1/TLR4/ NF-κβ TRAF6	↓ Apoptosis, inflammation, fibrosis	[74,166]
BMSCs HUCMSCs MSCs	HL-1 H9c2 NRCMs	MI (LAD)	miR-19a/b miR-19	SOX6/AKT JNK3/Casp3 PTEN/AKT	↓ Fibrosis, CM apoptosis	[167,168]
MSCs	-	MI (LAD)	miR-590-3p	Hoxp, Homer1,Cdk1,Cdk8	↑ CM proliferation	[169]
MSCs	CMECs	MI (LAD)	miR-543	Col4a1	↑ Proliferation, angiogenesis	[170]
CPCs	HUVECs	MI (LAD)	miR-322	NOX2	↑ Angiogenesis	[171]
BMSCs	-	MI (LAD)	miR-301	-	↓ Autophagy	[172]
BMSCs	CMs	MI (LAD)	miR-183-5p	FOXO1	↓ Apoptosis, oxidative stress	[173]
BMSCs HUCMSCs	NRCMs HUVECs	MI (LAD)	miR-133 miR-133a-3p	Sanil1 AKT	↑ Angiogenesis ↓ Inflammation, fibrosis, apoptosis	[130,174]
Plasma	H9c2	MI (LAD)	miR-342-3p	SOX6 TFEB	↓ Apoptosis, autophagy	[105]
EPCs	-	MI (LCA)	miR-218-5p miR-363-3p	p53/JMY	↓ Fibrosis	[175]
BMSCs	H9c2	MI (LAD)	miR-455-3p	MEKK1/MEKK4/ JNK	↓ Apoptosis	[176]
BMSCs MSCs	NMVM HUVECs RAW264.7	MI (LCA, LAD)	miR-182-5p miR-182	TLR4/NF-κβ GSDMD	↑ M1 → M2 polarization ↓ Inflammation, pyroptosis	[177,178,179]
Plasma	HUVECs HEK293Ts	Carotid artery injury	miR-193a-5p	ACVR1	↓ Oxidative stress	[180]
BMSCs cCFU-Fs	H9c2 HUVECs	MI (LAD)	miR-221-3p	PTEN/AKT	↑ Angiogenesis ↓ Fibrosis, apotosis	[142,181]
HUCMSCs hiPSC-Ecs	Ac16 hiPSC-CMs	MI (LAD)	miR-100-5p	FOXO3/NLRP3 PP-1β/SERCA-2a	↑ Ca^+2^ homeostasis ↓ Inflammation, pyroptosis	[182,183]
BMSCs	CMs	MI (LAD)	miR-22	Mecp2	↓ Apoptosis, fibrosis	[184]
BMSCs	NRCFs	MI (LAD)	miR-212-5p	NLRC5/VEGF/TGF-β/SMAD	↓ Fibrosis	[185]
HUCMSCs	-	MI (LAD)	miR-200b-3p	BCL2L11/NLRP1	↓ Apoptosis, inflammation	[186]
HUCMSCs	HUVECs	MI (LAD)	miR-423-5P	EFNA3	↑ Angiogenesis, migration ↓ Fibrosis	[187]
HUCMSCs	HUVECs H9c2	MI (LCA)	miR-223	p53/S100A9	↑ Angiogenesis ↓ Fibrosis	[128]
M2 macrophage	HL-1	MI (LAD)	miR-378a-3p	ELAV1/NLRP3/ Caspase1/ GSDMD	↓ Pyroptosis	[188]
MSCs	NMCMs	MI (LAD)	miR-150-5p	TXNIP	↓ Apoptosis	[189]
Plasma	H9c2 HEK293T	MI (LAD)	miR-130a-3p	ATG16L1	↓ Inflammation, autophagy	[190]
ADSCs	NMCMs	MI (LCA)	miR-224-5p	TXNIP	↓ Apoptosis, necrosis	[191]
iPSC derived CMs	iCMs	MI (LAD)	miR-106a-363 (cluster)	Notch3	↑ Proliferation	[192]
MSCs	NRCMs NRCFs	MI (LAD)	miR-4732	-	↑ Angiogenesis ↓ Fibrosis, apotosis	[193]
HUCMSCs	CMs	MI (LAD)	miR-23a-3p	DMT1	↓ Ferroptosis	[129]
mESs MEFs	H9c2 HUVECs	MI (LAD)	miR-294	-	↑ Angiogenesis, proliferation	[194]
**lncRNAs**						
**Source of Exosomes**	**In Vitro** **Essay**	**In Vivo Model**	**lncRNA**	**Signaling**	**Biological Effects**	**Refs.**
Plasma	HUVECs HMVECs	MI (LAD)	KLF3-AS1	miR-138-5p	↓ Apoptosis, pyroptosis	[195]
Cardiac myocites hPSC-CVPC	NRCMs HUVECs	MI (LAD)	MALAT1	miR-92a/KLF2 miR-497	↑ Angiogenesis	[196,197]
ECs	CMs	MI (LAD)	LINC00174	SRSF1/p53	↓ Apoptosis, autophagy	[198]
CMs	CFs	MI (LAD)	AK139128	-	↓ Fibrosis	[199]
MSCs	H9c2	MI (LAD	UCA1	miR-873/XIAP	↓ Apoptosis	[200]
MSCs	-	MI	TARID	Tcf21/Smad3/TGF-β	↓ Fibrosis	[201]
**circRNAs**						
**Source of Exosomes**	**In Vitro** **Essay**	**In Vivo Model**	**lncRNA**	**Signaling**	**Biological Effects**	**Refs.**
BMSCs		MI (LAD)	circ_002113	miR-188-3p/RUNX1	↓ Apoptosis	[202]
HUCMSCs	H9c2	MI (LAD)	circ_0001273	-	↓ Apoptosis	[203]
CMs Ac16	ECs Ac16	MI (LAD)	circ_HIPK3	miR-29a/VEGFA miR-33a-5p/IRS1	↑ Angiogenesis ↓ Apoptosis	[89,90]
ADSCs	HL-1	-	circ_0001747	miR-199b/MCL1	↑ Proliferation, viability ↓ Inflammation, apoptosis	[204]
ECs	CMs	MI (LAD)	circWhsc1	Trim59/Stat3/Cyclin B2	↑ Proliferation ↓ Fibrosis	[205]
Plasma	VSMCs CMECs	MI (LAD)	circCEBPZOS	miR-1178-3p/PDPK1	↑ Angiogenesis	[206]

## 3. Small EVs in the Pathological Process of Cardiomyopathies

Cardiomyopathies are defined as “a myocardial disorders in which the heart muscle is structurally and functionally abnormal, in the absence of coronary artery disease, hypertension, valvular disease, and congenital heart disease, however, sufficient to cause the observed myocardial abnormality” [207]. Cardiomyopathies can be classified into primaries or secondaries. Primary cardiomyopathies are mostly idiopathic, leading to heart failure and sudden death, while secondary cardiomyopathies develop in response to several extrinsic factors such as hypertension, metabolic disorders, drug-induced myopathy, ischemic heart disease and coronary artery disease [208,209,210].

### 3.1. Hypertrophic Cardiomyopathy

Hypertrophic cardiomyopathy (HCM) is one of the most common cardiac genetic conditions, with a prevalence greater than 1 in 500 in the general adult population [211]. This inherited disorder is characterized by left ventricular hypertrophy (>15 mm for adults), that cannot be only attributed to abnormal load conditions [212] (Figure 2D). Essential histopathological features include myocyte hypertrophy and disarray alongside heightened myocardial fibrosis; the combination of these hallmarks leads to left ventricular outflow track obstruction, impaired diastolic function and cardiac arrhythmias [213]. Genetically, HCM is an autosomal-dominant disorder caused by mutations in genes encoding for contractile and structural proteins of the cardiac muscle sarcomere apparatus [214]. Genetic analysis has improved our knowledge about the molecular bases of HCM, enabling clinicians to make an early identification prior to the onset of cardiac disease.

#### 3.1.1. Small Extracellular Vesicle Transcriptomic Analyses in Hypertrophic Cardiomyopathy

In this pathological scenario, James et al. (2021) [215] performed a transcriptomic analysis on small EVs derived from human-induced pluripotent stem cell-derived cardiomyocytes (hiPSC-CMs) with or without the c.ACTC1^G301A^ mutation. This model of HCM [216] was chosen for this study as it had previously been shown to recapitulate many key disease phenotypes including abnormal contractility, Ca^2+^ sensitivity/handling, arrhythmogenesis and hypertrophic brain natriuretic peptide signaling. Transcriptomic analysis of HCM small EVs has shown that CMs alter their EV cargo when HCM sarcomeric mutations are present. To be more precise, they observed differences in snoRNA cargo within HCM-released small EVs that specifically is altered when HCM hiPSC-CMs were subjected to an increased workload. In total, 12 snoRNAs were identified including 10 SNORDs (SNORD6, SNOTRD116-23, SNORD116-25, SNORD116-29, SNORD18A, SNORD42A, SNORD43, SNORD58C, SNORD60, and SNORD 101) and 2 SNORAs (SNORA3B and SNORA20). The functional role of these snoRNAs is related with post-translational modifications and alternative splicing processes differentially regulated in HCM (Table 3).

#### 3.1.2. Mechanistic Insights into Small Extracellular Vesicle Related with Hypertrophic Cardiomyopathy

Some years ago, Tian et al. (2018) demonstrated an upregulation of miRNA-27a levels in both infarcted myocardial tissue and systemic circulation in a rodent model of MI. The identified miRNA-27a exhibited a propensity for incorporation into small EVs, contributing to oxidative stress and promoting hypertrophic gene expression via modulation of the Nuclear factor (erythroid-derived 2)-like 2/Kelch-like ECH-associated protein 1 (Nrf2/keap1) signaling pathway [217]. Moreover, clinical studies corroborated increased miRNA-27a levels in failing hearts and systemic circulation, suggesting its potential utility as a diagnostic and prognostic biomarker for HF [218,219,220]. Furthermore, the same lab evidenced that miR-27a* exhibited resistance to degradation and mirrored the expression pattern of miRNA-27a in chronic HF. This miR-27a* is packaged into small EVs and taken up by CM-targeting Z-line-associated protein PDLIM5, thereby contributing to hypertrophic gene expression [221].

#### 3.1.3. Therapeutic Approaches

A recent study has evidenced that YF1, a derived non-coding RNA from cardiosphere-derived cell exosomes alleviates cardiomyocyte hypertrophy, inflammation, and fibrosis associated with HCM in transgenic mice harboring a clinically relevant mutation in cardiac troponin I (cTnIGly146) [222] (Figure 2D).

**Table 3 cells-13-00265-t003:** Summary of ncRNAs associated with cardiomyopathies (↑ upregulated and ↓ downregulated).

Assay	DE ncRNAs	Main ncRNAs	Sample	Ref.
**Hypertrophic cardiomyopathy >> snoRNAs**				
James et al. (2021)	12 snoRNA	SNORD6, SNOTRD116-23, SNORD116-25, SNORD116-29, SNORD18A, SNORD42A, SNORD43, SNORD58C, SNORD60, SNORD 101, SNORA3B and SNORA20)	hiPSC-CMs	[215]
**Dilated cardiomyopathy >> miRNAs**				
Zhang et al. (2023)	↑ 48 miRNAs ↓ 44 miRNAs	miR-423-5p, hsa-miR-185-5p, hsa-miR-150-5p, hsa-miR-10a-5p_R-1, hsa-miR-1304-3p_1ss13CA, sa-miR-3138_L-5R+2	Plasma (DCM patient)	[223]

### 3.2. Dilated Cardiomyopathy

Dilated cardiomyopathy (DCM), with an incidence of 1 in 2500 individuals, is defined as the presence of left ventricular dilation along with systolic dysfunction [208,224]. Moreover, DCM is frequently associated with an increased likelihood of severe arrhythmias, which suggests the pathological affection of the cardiac conducting system. Finally, with disease progression, the right ventricle and diastolic function are affected, leading to HF and death [225] (Figure 2E). Some genes are associated with the initiation, progression and pathology of DCM; nonetheless, while these genes seem to be linked to DCM, only a limited number directly contribute to the onset of DCM owing to genetic variations [226,227].

#### 3.2.1. Small Extracellular Vesicle Transcriptomic Analyses in Dilated Cardiomyopathy

Zhang et al. (2023) [223] performed high-throughput sequencing in plasma exosomes of DCM patients with chronic heart failure (CHF) and healthy controls, and a total of 3687 miRNAs were detected in these biological samples. However, only 92 miRNAs were significantly differentially expressed between the two groups; 48 miRNAs were upregulated and 44 miRNAs were downregulated [223]. Six of these miRNAs have been identified as significant contributors to the development of DCM through diverse mechanisms, such as the regulation of fibrosis (miR-423-5p, hsa-miR-185-5p, hsa-miR-150-5p, hsa-miR-10a-5p_R-1) [228,229,230,231,232,233], hypertrophy (hsa-miR-150-5p) [231,232], inflammation (hsa-miR-1304-3p_1ss13CA, hsa-miR-150-5p) [231,232,234], oxidative stress (hsa-miR-1304-3p_1ss13CA) [234], angiogenesis (hsa-miR-150-5p) [235] and mitochondrial function (sa-miR-3138_L-5R+2) [236] (Table 3).

#### 3.2.2. Small Extracellular Vesicle Proteomic Analyses in Dilated Cardiomyopathy

Bayes-Genis laboratory explored the proteomic signature of plasma-derived small EVs obtained from DCM patients and healthy controls. A total of 176 proteins (74.6%) were shared by controls and DCM patients, whereas 51 proteins were exclusive for the DCM group and 7 proteins were exclusive for the control group [237]. They observed that some proteins were generally over-represented in the cargo proteome of circulating DCM small EVs compared with control small EVs. These included fibrinogen, crucially associated with a high risk of cardiovascular disease due to its contribution to endothelial injury, plasma viscosity and thrombus formation [237,238,239,240,241,242]; serotransferrin, related with anemia as comorbidity in HF patients [237,243,244,245,246,247]; protease inhibitor α-1-antitrypsin (AAT) as a putative biomarker for the evaluation of disease status [248,249]; and several apolipoproteins [237]. Gene ontology analysis evidenced that proteins associated with stress as well as with protein activation were found to be more abundant in DCM small EVs when compared to control samples [237].

#### 3.2.3. Mechanistic Insights into Small Extracellular Vesicle Related with Dilated Cardiomyopathy

Wu et al. (2018) [250] analyzed three different serum exosomal miRNAs, exo-miR-92b-5p, exo-miR-192-5p and exo-miR-320a, in patients with DCM and acute HF (AHF) vs. healthy volunteers. In the study, exo-miR-92b-5p was increased in DCM-AHF patients compared to control, and was finally considered as a potential biomarker that potentially predicts DCM-AHF in patients [250]. Recent research with angiotensin II-stimulated hiPSCs differentiated cardiomyocytes has evidenced that miR-218-5p is upregulated in the DCM-Exos. This microRNA has been identified as a critical contributor to fibrogenesis through the activation of Tgf-β signaling after the suppression of TNFAIP3 [251].

#### 3.2.4. Therapeutic Approaches

Several labs have reported the therapeutic role of small EVs in DCM. Vandergriff et al. (2015) [252] analyzed the therapeutic role of cardiac stem cell-derived exosomes (CSC-exo) in a mouse model of doxorubicin-induced DCM. Systemic delivery of human CSC-exo in mice showed improved heart function via echocardiography, as well as decreased apoptosis and fibrosis [252]. Sun et al. (2018) [253] proved that mesenchymal stem cell-derived exosomes (MSC-Exos) alleviate inflammatory cardiomyopathy by improving the inflammatory microenvironment of the myocardium, especially by regulating the activity of macrophages in a mouse model of DCM [253]. Ni et al. (2020) [254] evidenced that trophoblast stem cell-derived exosomes (TSC-exos) could alleviate DOX-induced cardiac injury via the let-7i/YAP pathway. They observed an improvement of cardiac function and decreased inflammatory responses, accompanied by downregulated YAP signaling [254] (Figure 2E). Zhang et al. (2022) [255] evidenced that small EVs derived from KLF2-overexpressing endothelial cells reduced cardiac inflammation and ameliorated left ventricular dysfunction in DCM mice by targeting the CCR2 protein to inhibit Ly6Chigh monocyte mobilization from the bone marrow [255].

### 3.3. Diabetic Cardiomyopathy

The prevalence of diabetes mellitus (DM) is approximately 9.3% of the world population [256]. In this vast group, HF has emerged as the most common cardiovascular complication of diabetes [257]. Diabetic cardiomyopathy (DmCM) is a myocardial-specific complication that is associated with coronary microvascular dysfunction and increases the risk of HF in patients with diabetes [258] (Figure 2F). DmCM is characterized by left ventricle dysfunction, CM apoptosis and interstitial fibrosis developed in the absence of coronary artery disease, valvular disease and/or hypertension [259,260].

#### 3.3.1. Mechanistic Insights into Small Extracellular Vesicle Related with Diabetic Cardiomyopathy

Some years ago, Gonzalo-Calvo et al. (2017) [261] evidenced that circulating miR-1 and miR-133a levels are actively released from CM exosomes in response to lipid overload and are robustly associated with myocardial steatosis in type 2 diabetes patients [261]. Moreover, microRNAs which are encapsulated within exosomes offer a stable source of information to study the role of miRs associated with HF in diabetic hearts with preserved ejection fraction (HFpEF). Huang et al. (2022) [262] evidenced the association of exosomal miR-30d-5p and miR-126a-5p with diabetic HFpEF [262]. It has been shown that circulating miR-30d downregulation reduces the cardioprotective role of miR-30d in HF [263,264,265], whereas miR-126a downregulation decreases cardiac microvessel density and impairs ventricular function [266].

In addition, recent research evidenced that exosomes deliver Mst1 protein between cardiac microvascular endothelial cells (CMECs) and CM, playing a pivotal role in the development of DmCM. In this scenario, the increase in Mst1 in CM inhibits cell autophagy, enhancing the apoptotic CM ratio and, moreover, affecting glucose metabolism which leads to insulin resistance that finally contributes to DmCM and impaired cardiac function [267]. The same lab has recently demonstrated that exosomes mediate the interaction between CMECs and CFs. They observed that exosomes derived from CMECs under high glucose were rich in TGF-β1 mRNA, which significantly promoted the activation of CFs. This condition aggravates perivascular and interstitial fibrosis in mice with DmCM [268].

#### 3.3.2. Therapeutic Approaches

Heat shock protein (Hsp) response is a cellular intrinsic defense mechanism [269]. The expression of these proteins in type 1 and type 2 diabetes are decreased, contributing to diabetes-induced organ damage [270]. In this scenario, it has been evidenced that CM exosomes derived from a transgenic mouse model with cardiac-specific overexpression of Hsp20 protected against in vitro hyperglycemia-triggered cell death, as well as in vivo STZ-induced cardiac adverse remodeling. Thus, Hsp20-engineered exosomes might be a novel therapeutic agent for DmCM (Figure 2D). Moreover, Lin et al. (2019) [271] evidenced that mesenchymal stem cell (MSC)-derived exosomes significantly increased the levels of fatty acid transporters (FATPs) and fatty acid beta oxidase (FA-β-oxidase), whereas TGF-β1 and Smad2 mRNAs levels were significantly reduced. Such molecular regulation indicates that MSC-derived exosomes improve DM-induced myocardial injury and fibrosis via inhibition of the TGF-β1/Smad2 signaling pathway [271]. Similarly, parasympathetic ganglionic neuron-derived exosomes (PGN-exos) are able to inhibit apoptosis, improve cell viability and restore levels of anti-apoptotic protein Bcl-2 in diabetes-induced H9c2 cells [272]. Finally, recent data support the notion that ginsenoside RG1 (RG1)-induced MSCs secrete exosomes that can alleviate DmCM. Mechanistically, exosomes derived from RG1-induced MSCs transferred circNOTCH1 into macrophages, activating the NOTCH signaling pathway through the regulatory axis consisting of circNOTCH1, miR-495-3p and NOTCH1 [273] (Figure 2F). All these findings may contribute to the development of new therapeutic approaches for DmCM.

## 4. Small EVs in the Pathological Process of Atrial Fibrillation

Atrial fibrillation (AF) is the most common electrical disorder in humans. With a 2% prevalence in the general population, the incidence of AF raises to almost 10% in the elderly (+80 y). AF substantially contributes to morbidity and mortality by significantly altering the quality of life and, moreover, increasing the risk of embolic stroke and HF [274,275]. The risk of developing AF is enhanced by distinct cardiac and medical conditions, such as hypertension, cardiomyopathy, valvular dysfunction or obstructive sleep apnea [276,277].

AF is characterized by an irregular electrical pattern in the atrial chambers, lacking the P wave in the ECG (Figure 2C). Clinically, AF can be classified according its temporal duration in three distinct types; paroxysmal, persistent and permanent AF [276,277]. AF is generally considered as paroxysmal AF when the fibrillatory episodes self-terminate within seven days. Paroxysmal AF may progress to persistent and finally chronic or permanent states that fail to self-terminate. In the last decade, an increasing number of studies have reported the contribution of extracellular vesicles to the pathophysiology of AF. In the following paragraphs, we will summarize the current state-of-the-art information regarding the transcriptomic and proteomic EV analyses in different AF conditions, as well as an array of studies in which the role of discrete molecules is studied. Finally, the first insights of a therapeutic approach to revert AF conditions using EV therapy are envisioned.

### 4.1. Small Extracellular Vesicle Transcriptomic Analyses in Atrial Fibrillation

Several studies have investigated the concentration of small EVs in AF, providing evidence of an increased number of small EVs in AF vs. sinus rhythm controls [278,279]. Gene expression analyses have also been performed, most of them focusing on microRNA differential expression [280,281,282,283,284,285,286], while others demonstrated differentially lncRNA [287,288], circRNA [289] and protein [290,291,292] loading. Four different studies analyzed the differential expression of microRNAs in AF vs. non-AF patients. Siwaponanan et al. (2022) [280] revealed that nineteen microRNAs were significantly higher in AF vs. non-AF, and six were subsequently validated (miR-106b-3p, miR-590-5p, miR-339-3p, miR-378-3p, miR-328-3p and miR-532-3p). Zhu et al. (2022) [283] screened by high-throughput sequencing analysis and subsequently verified by qRT-PCR the differential expression of exosomal miRNAs, identifying that miR-124-3p, miR-378d, miR-2110 and miR-3180-3p were remarkably upregulated, while miR-223-5p, miR-574-3p, miR-125a-3p and miR-1299 were downregulated. Similarly, Wei et al. (2020) [284] searched for differences in the exosomal miRNAs between AF and normal sinus rhythm (SR) patients by combining high-throughput sequencing results and real-time PCR. A total of 20 microRNAs were initially identified, among which miR-92b-3p, miR-1306-5p and miR-let-7b-3p were differentially validated. While most of these authors highlight that these microRNAs can constitute promising biomarkers to assess AF in patients, there are limited mechanistic insights into the functional role of these microRNAs in AF pathophysiology, as only miR-124-3p has been reported to modulate the Wnt/β-catenin signaling pathway via AXIN1 in CFs [283] (Table 4).

Three additional studies have investigated the exosomal microRNA differential loading in patients in distinct phases of AF, i.e., paroxysmal, persistent and permanent AF. Mun et al. (2019) [285] analyzed exosomes from the serum of patients’ supraventricular tachycardia (SVT) as the controls, and paroxysmal AF and persistent AF patients through microRNA microarray analysis. Forty-five miRNAs were significantly higher in patients with persistent AF, but not in patients with paroxysmal AF as compared to control. Similarly, Wang et al. (2019) [282] identified 39 differentially expressed exosomal microRNAs from plasma of persistent AF and SR patient. Four of them were subsequently validated, i.e., miR-483-5p, miR-142-5p, miR-223-3p and miR-223-5p, and multivariable logistic analyses demonstrated that one of them, i.e., miR-483-5p, was independently correlated with AF. Finally, Hao et al. (2022) [281] also analyzed the differential expression of microRNAs in isolated exosomes from atrial myocytes and patient serum, and particularly focused on the contribution of miR-210 in AF, directly targeting GPD1L, thus regulating atrial fibrosis via the PI3K/AKT signaling pathway (Table 4). Interestingly, there are scarce microRNA similarities between the distinct studies, suggesting a lack of reproducibility. Such low reproducibility might be attributed to other parameters than AF per se, such as variations in sample collection (serum vs. plasma, systemic vs. intracardiac blood sampling) as well as differences in the microRNA platforms employed for microRNA differential discovery. It is important to note that such variations should be minimized during the validation steps, particularly when using RT-qPCR. Thus, protocol standardization would be desirable to further ascertain the global applicability of these microRNAs as AF biomarkers.

In addition, the exosomal microRNA cargo has also been investigated in AF-associated cardiovascular diseases, such as stroke. Xie et al. (2023) [286] conducted an analysis of serum exosomes from healthy individuals with SR, AF and AF-ischemic stroke patients by deep sequencing. They identified that miR-641 and miR-30e-5p were significantly upregulated in AF-ischemic stroke patients. These authors suggest that these microRNAs can be considered biomarkers of ischemic stroke in AF patients. However, no mechanistic insights are provided in regard to how these microRNAs are related to this CVD (Table 4).

Different studies have provided evidence that the deposition of epicardial adipose tissue (EAT) surrounding the heart is intimately linked with the onset of AF [293,294]. Furthermore, several studies have demonstrated that EAT can release signaling molecules that may contribute to the initiation of such electrophysiological disorders [295,296]. Therefore, different studies have investigated the EAT-exosomal cargo in AF vs. non-AF patients. These studies have placed particular emphasis on the differential expression of microRNAs [297], lncRNAs [287,288] and circRNAs [289]. While gene ontology analyses of the differentially expressed non-coding RNAs revealed functional categories, such as metabolism and stress response, which might contribute to the pathogenesis of AF, no further insights are reported about the mechanistic link between these non-coding RNAs and AF pathophysiology.

### 4.2. Small Extracellular Vesicle Proteomic Analyses in Atrial Fibrillation

To date, only three studies have investigated the protein content of small EVs in an AF context [290,291,292]. Ni et al. (2021) [292] identified differentially expressed exosomal proteins in AF and non-AF patients, highlighting that bioinformatic analyses revealed enrichment in proteins involved in anticoagulation, complement system and protein folding. Shaihov-Teper et al. (2021) [291] studied the role of EAT-derived small EVs in the pathogenesis of AF. By generating culture explants from patients with AF, these authors uncovered more secreted small EVs with greater amounts of proinflammatory and profibrotic cytokines in AF patients compared to those without AF. Mechanistically, they demonstrated that while EAT-derived small EVs from patients with and without AF shortened the action potential duration of induced pluripotent stem cell-derived CMs, only those from AF patients induced sustained reentry. Additionally, Weiss et al. (2021) [290] explored the differences in circulating small EV proteomic profiles in rivaroxaban-treated non-valvular AF patients as compared with matched warfarin controls. These authors demonstrated that circulating small EV profiles were fundamentally altered, with a decrease in highly pro-inflammatory protein expression and complement factors, along with increased expression of negative regulators of inflammatory pathways. Therefore, these data suggest the notion that differential protein cargo in small EVs is also relevant in the AF setting.

### 4.3. Mechanistic Insights into Small Extracellular Vesicle Related with Atrial Fibrillation

Different molecular cascades have been implicated as triggering factors of AF. In this context, the contribution of small EVs to AF has been reported to play a fundamental role in inducing fibrosis [298,299,300,301,302], inflammation [302,303,304,305], oxidative stress [302] and apoptosis [306]. Within the context of AF-atrial fibrosis, both microRNAs and lncRNAs have been identified. Exosomes containing miR-23a-3p, derived from CFs, can target SCL7A11, contributing to AF pathophysiology by increasing ferroptosis [299]. Additionally, exosomes containing lncRNA LINC00636 modulate miR-450a-2-3p and, thus, MAPK1, leading to improved cardiac fibrosis in AF [298]. Xu et al. (2022) [300] reported that exosomes, derived from MSCs, overexpressing Nrf2, inhibited cardiac fibrosis in AF, while Xu et al. (2021) implicated miR-324-3p-regulating Tgf-ß1 and, consequently, fibroblast proliferation in AF. In the context of inflammation, the lncRNA LRON was found to promote M2 macrophage polarization, from atrial myocytes, by delivering miR-23a, thus decreasing atrial fibrosis [304], while the exosomal-loaded lncRNA PVT1 regulates M1 macrophage polarization from angiotensin II-treated CMs [303]. Curiously, rapid atrial pacing modulates Kca3.1 function, leading to the promotion of proinflammatory exosome secretion with the activation of the Akt/RAb27a signaling pathways [305]. Importantly, Chen et al. (2021) [302] reported that EV containing MIAT can influence multiple triggering signals leading to AF, such as atrial fibrosis, inflammation and oxidative stress. This process is modulated by CXCL10 regulation via miR-485-5p. Finally, in the context of AF, CM apoptosis can also be modulated by the delivery of miR-148a exosomes derived from BMSCs, modulating SMOC2 expression [306].

Besides the biological processes involved in AF, additional evidence is emerging. For instance, Yan et al. (2021) [307] reported that lncRNA XIST, shuttled by adipose tissue-derived mesenchymal stem cell-derived small EVs, suppresses myocardial pyroptosis in AF by disrupting miR-214-3p-mediated Arl2 inhibition. In addition, Li et al. (2020) [308] revealed that myofibroblast-derived exosomes containing miR-21-3p were capable of modulating Cav1.2, thus contributing to electrical remodeling in AF. In sum, evidence is emerging on the functional role of discrete EV-mediated non-coding RNAs contributing to the pathophysiology of AF.

### 4.4. Therapeutic Approaches

The importance of small EVs in cell–cell communication in homeostasis and diseases is undoubtedly emerging, but most importantly, their therapeutic application is also emerging. Seminal work by Parent et al. (2023) [309] reported a randomized controlled trial in which the prevention of AF, after open-chest surgery, was evaluated, yielding promising results. In this context, injection of small EVs at the time of open-chest surgery shows prominent anti-inflammatory effects and effectively prevented AF due to sterile pericarditis (Figure 2C).

## 5. Small EVs in the Pathological Process of Heart Failure

Heart failure (HF) is a complex heterogeneous clinical syndrome, representing the terminal stage of several CVDs and associated with a high mortality. It affects approximately 1% to 2% of the adult population [310,311]. This syndrome is produced by an impairment of ventricular filing or an ejection of blood associated with symptoms of dyspnea, fatigue and in some cases, pulmonary edema, increased sympathetic activity and circulation redistribution [310,312] (Figure 2B). The main risk of HF is its constant progression. Therefore, the exploration of biomarkers for early diagnosis and identification of potential therapeutic targets is essential in the ongoing effort to fight against HF. The discovery of small EVs, including exosomes, gives us new opportunities to identify diagnostic and therapeutic biomarkers [313,314]. In the following paragraphs, we summarize the potential of small EVs in HF as diagnosis biomarkers and their use in therapy.

### 5.1. Small Extracellular Vesicle Transcriptomic Analyses in Heart Failure

It has been widely documented that the number of small EVs is increased in the serum of patients with HF. The origin of these small EVs is not exclusive from the damaged myocardium, since hallmarks present on the surface of the small EVs reveal that they are also derived from ECs as well as from immune cells, contributing to the development of this syndrome, promoting the inflammatory process [315,316,317,318,319,320]. Small EVs’ cargo can thus be a source of HF diagnostic biomarkers. As previously said, many cardiac diseases end in HF; one of these is DCM. In order to find biomarkers for DCM diagnosis, Zhang et al. (2023) performed isolation of exosomes in patients with chronic heart failure (CHF) caused by DCM. By next generation sequencing, they identified 92 microRNAs differentially expressed, and gene ontology analysis revealed that these microRNAs are related to oxytocin signalling, Hippo signalling and Ras signalling, pointing them out as candidates for CHF diagnosis [223]. As mentioned in previous paragraphs, diabetes mellitus is related with complications in HF with preserved ejection fraction (HFpEF). In a diabetic rat model with HFpEF, circulating exosomes decreased their miR-30d-5p and miR-126a-5p expression levels, suggesting that they can be used as biomarkers for HF [262]. In this scenario, other authors proposed several microRNAs presents in exosomes from patients with HF as biomarkers for diagnosis such as miR-27a, miR-34a, miR-92b, miR-146, miR-194, miR-425 and miR-744 [250,321,322,323,324] (Table 5). Other biomarkers for HF are piRNAs, which are ncRNAs 24-32 nucleotides in length associated with PIWI proteins, repressing transposable elements, and thus keeping the integrity of the germinal cell line. The analysis of piRNA cargo in exosomes from serum in patients with HF by RNAseq identified 585 upregulated piRNAs and 4623 downregulated piRNAs, highlighting hsa-piR-02009 and hsa-piR-006426 as the most downregulated piRNAs, suggesting them as potential HF biomarkers [325] (Table 5). Following ncRNAs, circRNAs can be a promising source of biomarkers. Han et al. (2020) performed a screening of differentially expressed cirRNAs in HF-EXO by next generation sequencing, identifying 56 differentially expressed circRNAs, further suggesting that hsa-circ-0097435 can be used as a biomarker involved in myocardial cell injury [326] (Table 5).

### 5.2. Small Extracellular Vesicle Proteomic Analyses in Heart Failure

The study of proteomic signatures in small EVs from the plasma of patients with HF is also a significant tool for HF diagnosis. As HF is associated with others pathologies, a proteomic study performed by the isolation of small EVs from the plama of patients with HF, dyspnea and renal dysfunction shows that these small EVs bear Cystatin C and CD14 that can be used as biomarkers for diagnosis [329].

### 5.3. Mechanistic Insights into Small Extracellular Vesicle Related with Heart Failure

As previously pointed out, the great challenge in HF is its unstoppable progression, and in this progression, small EVs have an important role, controlling the cell-to-cell communication participating in several processes such as cell adhesion, apoptosis, immune response and vascular function [319]. miR-22-3p was previously known as an HF biomarker present in exosomes [327]. Functionally, miR-22-3p targets FURIN, and thus induces the expression of apoptotic-related genes in HF [330]. Transforming growth factor beta (TGF-β) is a cytokine upregulated in several cardiac diseases [331]. The isolation of exosomes from fibroblasts treated with TGF-β identified 50 genes differentially expressed related with cardiac hypertrophy. Cardiomyocytes treated with these exosomes expressed 40 of these differentially expressed genes, inducing an HF phenotype, thus pointing out fibroblasts as targets for HF treatment [332]. Sympathetic hyperactivity also plays an important role in the progression of CHF. Inflammation in the rostral ventrolateral medulla (RMLV), a key region for sympathetic control, excites the activity of neurons and promotes an increase in sympathetic outflow. In a CHF rat model, the levels of circulating microRNAs in exosomes vs. microRNA levels in the RVLM were compared, and 59 DE-microRNAs and 5 overlapping microRNAs were identified. Three of these overlapping microRNAs were miR-214-3p that was upregulated in exosomes and in RVLM, while let7g-5p and let7i were downregulated in exosomes and in RVLM of CHF rats. In vitro studies in PC12 cells showed that miR-214-3p enhanced the inflammatory response, and let7g-5p and let7i-p reduced neuroinflamation, suggesting that the circulating exosomes enhanced the inflammatory response in the RVLM, contributing to sympathetic hyperactivity in CHF [328]. NF-E2-related factor 2 (NRF2) mediated sympathetic activation, and its expression was decreased in HF. Importantly, cardiac-EVs enriched with microRNAs mediate the cross talk between heart- and brain-targeting NRF2, thus producing an unbalance in the sympathetic outflow [333].

### 5.4. Therapeutic Approaches

Heart transplantation stands out as the most effective treatment against HF. However, its effectiveness is impaired by the reduced number of donors and the difficulty of organ preservation. Currently, cold static storage represents the main method of preservation, but its disadvantages limit the overall success of heart transplantation [334]. During cold ischemia preservation, the myocardial function of the transplanted heart is impaired and cell death increases. Of note, as mentioned in the previous paragraphs on therapeutic approaches, small EVs are deemed as promising therapeutic agents that could be used to avoid heart transplantation, and in this regard, several labs have conducted scientific studies in the context of HF. MSC-EVs therapy exerts a cardioprotective effect on the heart, improving its myocardial function and decreasing cell death [335]. Additional evidences reported that mesenchymal stem cell exosome (MSC-EXO) delivery produces a cardioprotective effect, promoting angiogenesis, and decreasing fibrosis and apoptosis by deactivating the Hippo-YAP pathway in HF [336,337]. The efficiency of MSC-EXO treatment can be improved with the administration of agomiR-125a-5p. Mechanistically, miR-125a-5p has KLF13, TGFB1 and DAAM1 as targets genes, involved in macrophages, CF and EC function, respectively. Moreover, agomiR-125a-5p treatment increased M2 macrophage polarization, promoted angiogenesis and attenuated fibroblast proliferation in a mouse and porcine model of I/R [338]. Treatment with BMSCs-EVs improved cardiac function and angiogenesis and alleviated fibrosis and inflammation in a rat model with acute myocardial infarction-induced HF via the delivery of BMP2 [339] (Figure 2). Exosomes derived from embryonic stem cells (ESCs-EXOs) are also able to induce myocardial angiogenesis in a transverse aortic constriction HF model through the FGF2 signalling pathway by attenuating myocardial damage [340]. Small EVs secreted by induced pluripotent stem cell-derived cardiovascular progenitors have a cardioprotective effect, making them a potential treatment for CHF with their cargo enriched in 16 microRNAs associated with tissue repair pathways [341]. Another source of small EVs suitable for HF treatment are small EVs from umbilical cord mesenchymal stem cells (HucMSC-EVs); these small EVs reduce oxidative stress and cell apoptosis through miR-100-5p, which has NOX4 as a target [342]. Small EVs from ECs with Krüppel-like factor 2 (KLF2) overexpression can be used as a treatment against DCM-HF; KLF2 exerts an anti-inflammatory effect, ameliorating left ventricular dysfunction in DCM mice. KLF2 targets CCR2 protein, preventing monocyte mobilization from the bone marrow [255]. Human trophoblast stem cell-derived exosomes (TSC-Exos) showed cardioprotective properties in a mice model of HF induced by doxycycline; the administration of TSC-Exos and miR-200b inhibitor decreases CM apoptosis, maintains the integrity of the mitochondria and improves cardiac function through an increase in Zeb1, which is a target of miR-200b and has an antiapoptotic effect [343,344] (Figure 2B).

In summary, HF is a complex syndrome with a difficult treatment process, and small EVs play a role in the development of the disease but can also be used as therapeutic or diagnosis tools.

## 6. Conclusions and Perspectives

Cell-to-cell communication represents a key biological process in cellular and tissue homeostasis that is mediated by different molecular mechanisms (see recent reviews [1,2,3]). In recent years, exosome-mediated cell-to-cell communication has emerged as a fundamental process contributing to cellular homeostasis. Multiple lines of evidence have also been reported, indicating that such exosome-mediated cellular communication is altered and impaired in pathological conditions, particularly in oncogenesis [4,11]. More recently, impaired exosome signaling has also been reported in distinct cardiovascular pathological contexts, including ischemic [67,68,69], electrical [280,281,282,283,284,285,286] and structural [215,223] cardiopathies. It is important to highlight in this context that increasing evidence is reported about the functional role of exosomes in distinct ischemic diseases, such as MI and coronary artery diseases, as well as in structural heart diseases such as DCM and HCM, as reviewed in this study. However, our understanding of the functional role of exosomes in electrical cardiovascular alterations is mostly confined to AF, while scarce or no evidence is yet reported in other relevant electrophysiological defects, such as Brugada, LQT and SQT syndromes, respectively. This might be indeed caused by the lower prevalence of these cardiac defects as compared to cardiac structural defects. In the next coming years, we will, therefore, witness increasing evidence of the functional role of exosomes in other cardiovascular diseases, such as valvular heart diseases and electrical cardiac wiring defects.

Exosome cargo selectively includes a large array of distinct biologically active molecules, such as lipids, proteins, coding and non-coding RNAs (microRNAs, lncRNAs and circRNAs) [1,2,3]. Evidence of the differential and selective distribution of combinations of these molecules, derived from distinct cardiovascular cell sources, i.e., cardiomyocytes, endothelial and/or epicardial cells and infiltrated immune cells, has been largely documented [298,299,300,301,302,303,304,305]. Such differential distribution, particularly of microRNAs, lncRNAs and circRNAs, in distinct pathological conditions has provided molecular hallmarks that can be exploited as sensitive biomarkers for the early diagnosis and prediction of distinct cardiovascular pathological conditions. Curiously, while our current identification of these biomarkers is greatly increasing, only very few of them are consistently validated in different studies. Such discrepancies might be derived, among other causes, from the variability of the exosome sample collection (plasma, serum, biopsies), their subsequent isolation procedures (ultracentrifugation, column binding) and their discovery/validation strategies (microarrays, next generation sequencing). Thus, there is an urgent need to invest in providing standardized pipelines that can pave the path for the discovery of robust biomarkers that can speed up their use into the clinical arena.

Curiously, while numerous reports highlight the fitness of distinct exosome-contained non-coding RNAs as cardiovascular disease biomarkers, the molecular mechanisms underlying their contribution to such pathology are, in most cases, scarce or even absent. Therefore, additional studies should be performed to increase our understanding of these molecular pathways. Such information will pave the path to design novel strategies by which exosome cargos can be modified, thus enabling the customization of cell–cell communication pathway modulation. Such cellular and molecular approaches will allow, in the coming future, the generation of personalized exosome cargos, which will selectively modulate specific signaling pathway responses, therefore opening novel therapeutic roads. As mentioned previously, exosomes are secreted by most eukaryotic cells, including embryonic stem cells (ESC). These ESC exosomes represent an innovative cell-free approach to harnessing the robust regenerative capabilities of ESC without the inherent risks associated with direct transplantation of ESC, such as the potential for teratoma formation [194,345]. Finally, while exosomes hold great promise for clinical applications, some associated challenges need to be overcome. In this context, it is crucial to emphasize the significance of the exosome delivery route, as it directly influences the therapeutic effectiveness of exosomes. Different administration methods for exosome delivery exist, ranging from less invasive techniques such as intranasal, inhalation and oral administration, to more invasive approaches like intravenous, intraperitoneal, intramyocardial and intraventricular injection. Numerous researchers suggest that intramyocardial injection stands out as the optimal delivery method for cardiac therapy. This is attributed to its ability to achieve the highest concentration of exosomes in the cardiac muscle, thereby preventing excessive accumulation in other organs [194,346,347,348]. The determination of optimal dosages for exosomes treatments is challenging, primarily influenced by delivery methodologies, and further by exosomes’ inherent characteristics such as their short half-life and Z-potential (aggregation index) [349]. Last but not least, for exosome treatments to transition into clinical practice, it is imperative to define biopharmaceutical safety standards and ensure cost-effectiveness.

To conclude, exosome-mediated cell-to-cell communication holds promise in understanding and treating cardiac diseases, offering potential biomarkers for early diagnosis. However, overcoming challenges related to standardized collection methods, molecular understanding, and therapeutic optimization is crucial for the successful clinical integration of exosome cardiac treatments.

## Figures and Tables

**Figure 1 cells-13-00265-f001:**
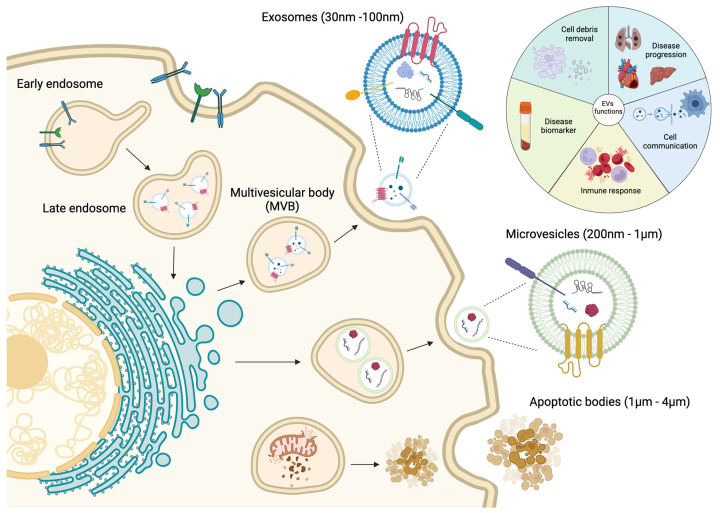
Schematic representation of the biological process of the genesis of extracellular vesicles, including the major classification due their size, content and functional impact.

**Figure 2 cells-13-00265-f002:**
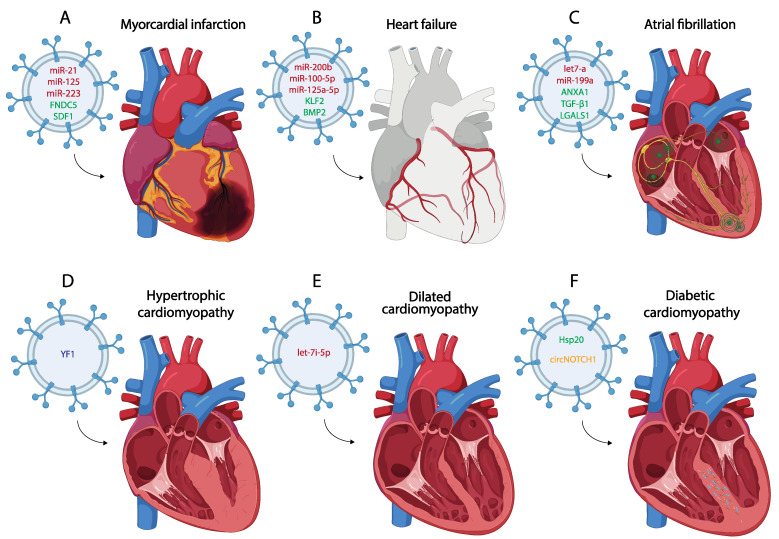
Schematic representation of the most representative small EV therapy in the treatment of different of myocardial infarction (**A**), heart failure (**B**), atrial fibrillation (**C**), hypertrophic cardiomyopathy (**D**), dilated cardiomyopathy (**E**) and diabetic cardiomyopathy (**F**). (miRNA red, protein green, lncRNA blue, circRNA yellow).

**Table 4 cells-13-00265-t004:** Summary of ncRNAs associated with atrial fibrillation (↑ upregulated and ↓ downregulated).

Study	DE ncRNAs	Main ncRNAs	Sample	Ref.
**microRNAs**				
Siwaponanan et al. (2022)	↑ 19 miRNAs	miR-106b-3p, miR-590-5p, miR-339-3p, miR-378-3p, miR-328-3p, and miR-532-3p	large EVs AF vs. non-AF	[280]
Zhu et al. (2022)	↑ 13 miRNAs ↓ 27 miRNAs	miR-124-3p, miR-378d, miR-2110, and miR-3180-3p miR-223-5p, miR-574-3p, miR-125a-3p, and miR-1299	plasma exosomes AF vs. non-AF	[283]
Wei et al. (2020)	↑ 33 miRNAs ↓ 117 miRNAs	miR-92b-3p, miR-1306-5p, let-7b-3p	plasma exosomes AF vs. non-AF	[284]
Mun et al. (2019)	↑ 45 miRNAs	miR-103a, miR-107,miR-320d, miR-486,let-7b	serum paroxysmal vs. persistent AF	[285]
Wang et al. (2019)	↑ 21 miRNAs ↓ 18 miRNAs	miR-483-5p, miR-142-5p, miR-223-3p	plasma exosomes AF vs. non-AF	[282]
Hao et al. (2022)	↑ 41 miRNAs ↓ 50 miRNAs	miR-210	atrial myocytes, serum exosomes	[281]
Xie et al. (2023)	↑ 31 miRNAs ↓ 37 miRNAs	miR-641 miR-30e-5p	plasma exosomes ischemic AF stroke	[286]

**Table 5 cells-13-00265-t005:** Summary of ncRNAs associated with heart failure (↑ upregulated and ↓ downregulated).

Assay	DE ncRNAs	Main ncRNAs	Sample	Ref.
**miRNAs**				
Zhang et al. (2023)	50 miRNAs ↑ 48 miRNAs ↓	miR-103a-3p miR-148a-3p	Plasma (CDM patient)	[223]
Huang et al. (2022)	2 miRNAs ↑	miR-30d-5p miR-126a-5p	HFpEF rat model	[262]
Xie et al. (2022)	miR-27a ↑	miR-27a	Serum (HF patient)	[324]
Beg et al. (2017)	2 miRNAs ↑	miR-486 miR-146a	Plasma (HF patient)	[323]
Wang et al. (2018)	2 miRNAs ↓	miR-425 miR-744	Plasma (HF patient)	[322]
Matsumoto et al. (2013)	2 miRNAs ↑	miR-194 miR-34a	Serum (Hf patient)	[321]
Wu et al. (2018)	miR-92b-5p ↑	miR-92b-5p	Serum (Hf patient)	[250]
Galluzzo et al. (2021)	20 miRNAs ↑ 12 miRNAs ↓	miR-22-3p	Plasma (HF patient)	[327]
Xiao et al. (2022)	5 miRNAs ↑ 13 miRNAs ↓	miR-214-3p let-7i-5p let-7g-5p	HF rat model	[328]
**piRNAs**				
Yang et al. (2018)	585 piRNAs ↑ 4623 piRNAs ↓	piR-02009 piR-006426	Serum (HF patient)	[325]
**circRNAs**				
Han et al. (2020)	29 circRNAs ↑ 27 circRNAs ↓	circ-0097435	Blood (HF patient)	[326]

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
