# Peer review of "Unraveling the Signaling Dynamics of Small Extracellular Vesicles in Cardiac Diseases"

_cells, 2024, doi:10.3390/cells13030265_

Round 1

Reviewer 1 Report

Comments and Suggestions for Authors

This review is up to date and the contents are comprehensive and inspirative to the revewer. I have a comment to the authors if they could have a touch of the delivery routes by including intravenous, intramyocardial, intracoronary, intrapericardial and cardiac patch etc to the therapeutic effects. Additionally, if back to the cardiac regenerative medicine, from where EV therapies be derived, how do the author think about the stem cell and the EVs in cardiac regeneration as discussed previously (https://doi.org/10.3390/cells10030641)?

Author Response

First of all, we would like to thank the reviewer her/his kind words regarding our manuscript about EVs and cardiac diseases. Following the reviewer's recommendation we have incorporated additional contents in the conclusion section. To address the suggestion about delivery routes of EVs for therapy purposes, we have also explored the potential of EVs derived from stem cells due to their ability to promote regeneration in a free-cell mode, thereby mitigating associated challenges. These modifications aim to enhance the comprehensiveness of our manuscript and align it more closely with the insightful recommendations from the reviewer.

Reviewer 2 Report

Comments and Suggestions for Authors

The review is informative and comprehensive 

My comments

- Figure 2 should be moved to the introduction to give a brief about the  cardiovascular diseases covered in the review

- A paragraph should be added about challenges and obstacles in exosomes application

Comments on the Quality of English Language

Minor editing of English language required

Author Response

First of all, we would like to thank the reviewer her/his kind words regarding our manuscript about EVs and cardiac diseases. In response to the reviewer's valuable recommendations, we have incorporated additional content in the conclusion section to specifically address the challenges and obstacles associated with the application of exosomes in therapy. Furthermore, we have made noteworthy enhancements to Figure 2, which provides a comprehensive overview of EVs therapy across various cardiac diseases. Although the figure has not been relocated to the introduction section, we have improved it by adding labels to each pathology. Additionally, we strategically referenced the figure throughout the text, particularly in the introductory sections of each pathology, to underscore its relevance. Finally, we have carefully gone through the whole manuscript editing English language. These modifications aim to enhance the comprehensiveness of our manuscript and align it more closely with the insightful recommendations from the reviewer.

Reviewer 3 Report

Comments and Suggestions for Authors

This manuscript extensively discussed the roles of extracellular vesicles (EVs) as diagnostics or therapeutics in multiple cardiovascular diseases, including myocardial infarction, dilated cardiomyopathy, hypertrophic cardiomyopathy, and diabetic cardiomyopathy. The authors summarized the regulation and roles of EV RNA cargo, especially miRNA, lncRNA, and circRNA, in cardiac disorders. Overall, this manuscript is of great significance and interest. Several concerns to be addressed:

1. EV definition: As a broad EV community consensus, EV has been defined into two main categories: exosomes and ectosomes (such as PMID: 36775986). Ectosomes cover all EVs budded from the plasma membrane. Microvesicles and apoptotic bodies belong to ectosomes. Please correct the description and introduction of EVs accordingly.

2. Disease types: The authors discussed most of the cardiac disorders, including both genetic and non-genetic. Not sure if the authors would like to include main vascular disorders, such as coronary artery diseases (see review PMID: 28495995). If not, the theme of this manuscript should be changed to heart diseases or similar, instead of general cardiovascular diseases. In addition, the authors also missed the EV studies in pathological cardiac hypertrophy (such as PMID: 32370947).

3.  EV RNA cargo: Although miRNA and lncRNA in EVs have been widely studied in CVDs, novel EV RNA species, such as circRNA, snoRNA, piRNA, and tDR/tsRNA, are also emerging. The authors did include circRNAs, snoRNA, and piRNA, but missed tsRNA/tDR. tsRNA/tDR seems to contribute to a significant portion of EV transcriptome and has been shown to be significantly regulated in response to cardiac ischemia in both cardiac cells and in bypass patients undergoing cardiac surgery (PMID: 35373532).

4. Use of term ‘exosome’: Even though exosome is defined as ‘derived from multivesicular bodies’, it is technically challenging to distinguish exosomes from small ectosomes. Therefore, I would recommend using small EVs instead of exosomes to describe most of the EV studies described in this manuscript, except there is clear evidence in the study demonstrating that their EVs are derived from multivesicular bodies.

Comments on the Quality of English Language

It is overall well written. Personally, I don't like the way to describe a study as ' Name et.al 20XX described/evidenced...'. However, it is not mandatory to change them.

Author Response

First of all, we would like to thank the reviewer her/his kind words regarding our manuscript about EVs and cardiac diseases.

Regarding several concerns to be addressed:

  1. EV definition: As a broad EV community consensus, EV has been defined into two main categories: exosomes and ectosomes (such as PMID: 36775986). Ectosomes cover all EVs budded from the plasma membrane. Microvesicles and apoptotic bodies belong to ectosomes. Please correct the description and introduction of EVs accordingly.

In response to the reviewer's valuable recommendations, we have incorporated additional information about exosomes and ectosomes in the introduction section.

  1. Disease types: The authors discussed most of the cardiac disorders, including both genetic and non-genetic. Not sure if the authors would like to include main vascular disorders, such as coronary artery diseases (see review PMID: 28495995). If not, the theme of this manuscript should be changed to heart diseases or similar, instead of general cardiovascular diseases. In addition, the authors also missed the EV studies in pathological cardiac hypertrophy (such as PMID: 32370947).

Thank you for your comments. In this regard, following the recommendation of the reviewer we have modified the title accordingly, using ‘cardiac disease’ instead of ‘cardiovascular diseases’. Moreover, we have introduced a new subheading with the mechanistic insight of miR-27a loaded into small EVs on the modulation of hypertrophic genes in cardiomyocytes.

  1. EV RNA cargo: Although miRNA and lncRNA in EVs have been widely studied in CVDs, novel EV RNA species, such as circRNA, snoRNA, piRNA, and tDR/tsRNA, are also emerging. The authors did include circRNAs, snoRNA, and piRNA, but missed tsRNA/tDR. tsRNA/tDR seems to contribute to a significant portion of EV transcriptome and has been shown to be significantly regulated in response to cardiac ischemia in both cardiac cells and in bypass patients undergoing cardiac surgery (PMID: 35373532).

Thank you for your valuable comments and suggestions. After a thorough review, it is important to acknowledge that the function and regulation of tRNAs in the context of various cardiac diseases have been documented; for instance, myocardial infarction (PMID: 36190649), cardiomyopathies (PMID: 36362661, PMID: 36736767, PMID: 34991096, PMID: 36553526), atrial fibrillation (PMID: 38190870), and heart failure (PMID: 37839439). However, while tRNAs have been identified in EVs within the plasma model of patients undergoing cardiopulmonary bypass and in in vitro assays, there is currently no evidence regarding the specific role of tRNAs in EVs related to the cardiac pathologies discussed in the review.

  1. Use of term ‘exosome’: Even though exosome is defined as ‘derived from multivesicular bodies’, it is technically challenging to distinguish exosomes from small ectosomes. Therefore, I would recommend using small EVs instead of exosomes to describe most of the EV studies described in this manuscript, except there is clear evidence in the study demonstrating that their EVs are derived from multivesicular bodies.

We appreciate the reviewer's observation, and we fully agree with her/his feedback. Throughout the entire manuscript-writing process, we exercised caution in addressing this aspect. Specifically, we consistently employed the term 'exosomes' when the original article referred to them as such, otherwise, we used the broader term 'EVs', which, in the revised version, has been refined to 'small EVs’.

All these modifications aim to enhance the comprehensiveness of our manuscript and align it more closely with the insightful recommendations from the reviewer.